# Vacillantins A and B, New Anthrone *C*-glycosides, and a New Dihydroisocoumarin Glucoside from *Aloe vacillans* and Its Antioxidant Activities

**DOI:** 10.3390/plants9121632

**Published:** 2020-11-24

**Authors:** Maram Al-Tamimi, Shaza M. Al-Massarani, Ali A. El-Gamal, Omer A. Basudan, Maged S. Abdel-Kader, Wael M. Abdel-Mageed

**Affiliations:** 1Department of Pharmacognosy, College of Pharmacy, King Saud University, PO. Box 2457, Riyadh 11451, Saudi Arabia; maltamimi1@ksu.edu.sa (M.A.-T.); salmassarani@ksu.edu.sa (S.M.A.-M.); basudan@ksu.edu.sa (O.A.B.); wabdelmageed@ksu.edu.sa (W.M.A.-M.); 2Department of Pharmacognosy, Faculty of Pharmacy, Mansoura University, El-Mansoura 35516, Egypt; 3Pharmacognosy Department, College of Pharmacy, Sattam Bin Abdulaziz University, Al-kharj 11942, Saudi Arabia; mpharm101@hotmail.com; 4Department of Pharmacognosy, College of Pharmacy, Alexandria University, Alexandria 21215, Egypt; 5Pharmacognosy Department, Faculty of Pharmacy, Assiut University, Assiut 71526, Egypt

**Keywords:** *Aloe vacillans*, Asphodelaceae, dihydroisocoumarin glucoside, anthraquinone, 9-anthrone *C*-glycoside, antioxidant activity

## Abstract

A new dihydroisocoumarin glucoside, vacillanoside (**3**), and two new anthrone *C*-glycosides microdantin derivatives; vacillantin A (**10**) and B (**11**), together with nine known compounds belonging to the anthraquinone, anthrone and isocoumarin groups were isolated from the leaves of *Aloe vacillans*. The structures were determined based on spectroscopic evidence including 1D and 2D nuclear magnetic resonance (NMR) spectroscopy and high resolution mass spectrometry (HRESIMS) data, along with comparisons to reported data. The leaves were used to extract compounds with different solvents. The extracts were tested for antioxidant activity with a variety of in vitro tests including 2,2-diphenyl-1-picrylhydrazyl (DPPH^•^), 2,2′-azino-bis (3-ethylbenzothiazoline-6-sulfonate (ABTS^•+^), ferric reducing antioxidant power assay (FRAP), superoxide, and nitric oxide radical scavenging assays. The dichloromethane fraction was most active, displaying significant free radical scavenging activity. The *n*-butanol fraction also showed notable activity in all assays. Therefore, these findings support the potential use of *A. vacillans* leaves as an antioxidant medication due to the presence of polyphenolic compounds.

## 1. Introduction

*Aloe* spp. are members of the bitter or Asphodelaceae family (previously known as Liliaceae). This family is represented by more than 600 species endemic to tropical and southern Africa, Madagascar, Jordan, the Arabian Peninsula, East Asian countries, and various islands in the Indian Ocean [1]. *Aloe* plants are used as traditional medicines and dietary supplements in several countries including Egypt, China, and India [2,3].

In Arabic, *Aloe* is known as “Alloeh”, which means “shiny substance with bitter taste”, in reference to its exudate [4]. *Aloe* spp., with a waxy surface on succulent leaves, are well-adapted to harsh climatic conditions with infrequent precipitation [5]. Traditionally, *Aloe* was used as a purgative and bowel cleansing agent, a blood purifier, a gargle for a sore throat, and externally to treat burns, venereal ulcers, and shingles [6]. The Greek Herbal of Dioscorides (41–68 AD) recommended oral use of *Aloe* spp. for constipation, and external application for the treatment of wounds, hemorrhoids, ulcers, and hair loss [3,5].

A vast number of reports are available on the biological activities of different extracts from *Aloe* spp. and isolated secondary metabolites. For example, anti-inflammatory [7], antioxidant [8], anti-aging [9], anti-diabetic [4], anticancer [10], and immunomodulatory [11] effects have been observed. *Aloe* is also commonly used in the food supplement industry for the management of obesity and hyperlipidemia [12]. Furthermore, in the cosmetics industry, *Aloe* gel is incorporated into many pharmaceutical preparations that are used externally such as cleansers, moisturizers, shampoos, lotions, and sunscreen products [5]. The internal use of this gel is regulated as a dietary supplement in the USA [13] and Europe [14]. An edible coating material made from *A. vera* gel increases the shelf-life of grapes and reduces the total microbial counts of stored products [15]. Different plant parts including the leaves, roots, and gels from various *Aloe* species have been thoroughly investigated, affording several classes of secondary metabolites including alkaloids, pre-anthraquinones, anthraquinones, anthrones, chromones, flavones, coumarin derivatives, and pyrones [16].

With approximately 24 reported species, the *Aloe* genus is considered one of the largest groups of succulent plants growing in the Kingdom of Saudi Arabia [17]. *Aloe vacillans*, Forssk. (Syn. *A. dhalensis* Lavrans, and *A. audhalica* Lavrans and hardy) grows on rocky mountain slopes in Yemen and Saudi Arabia at an altitude of approximately 8000 ft [18]. The plant is stemless, forming small rosette-shaped succulent leaves that show brown tooth margins at the base of the plant. Bright yellow to orange-red flowers are grouped in inflorescences [17].

The reported biological, therapeutic, and economic importance of the genus *Aloe* encouraged us to explore the chemical composition and potential biological activities of the constituents from the endemic species *A. vacillans* Forssk. (Appendix A) The present study deals with the isolation and identification of some constituents of *A. vacillans* growing in Saudi Arabia and the evaluation of the antioxidant and free radical scavenging activities of the extract and its different fractions including 2,2-diphenyl-1-picrylhydrazyl (DPPH^•^), 2,2′-azino-bis (3-ethylbenzothiazoline-6-sulfonate) (ABTS^•+^), ferric reducing antioxidant power assay (FRAP), and superoxide and nitric oxide radical scavenging assays.

## 2. Results and Discussion

### 2.1. Structure Elucidation of New Compounds

A phytochemical study of the CH_2_Cl_2_ and EtOAc soluble fractions of the leaves of *A. vacillans* using diverse chromatographic methods, afforded twelve compounds. The known compounds were identified as aloe-emodin (**1**) [19], feralolide (**2**) [20,21], 10-hydroxyaloins A and B (**4**, **5**) [22], aloin A and B (**6**, **7**) [19], microdontin A and B (**8**, **9**) [23], and elgonica-dimer A (**12**) [24,25] (Figure 1 and Appendix A) (Appendix A). The structures of the isolated compounds were identified based on a variety of spectroscopic techniques including 1D (^1^H, ^13^C, and DEPT-^13^C experiments) and 2D (^1^H-^1^H COSY, ^1^H-^13^C HSQC, ^1^H-^13^C HMBC, and ^1^H-^1^H NOESY) nuclear magnetic resonance (NMR) spectroscopy. Accurate mass measurements and comparisons with published data were also used (Appendix A), and electronic circular dichroism (ECD) experiments were conducted to determine the absolute configurations.

Compound (**3**) was isolated as a white amorphous powder. The spectral data of (**3**) including IR, UV, and NMR were very similar to that of feralolide (**2**) [20], suggesting a similar dihydorisocoumarin skeleton. HRESIMS showed quasi-molecular ion peaks at *m*/*z* 507.1500 [M+H]^+^ (calcd 507.1503 for C_24_H_27_O_12_), *m*/*z* 529.1320 [M+Na]^+^ (calcd 529.1322 for C_24_H_26_O_12_Na), and *m*/*z* 545.1069 [M+K]^+^ (calcd 545.1061 for C_24_H_26_O_12_K) with 162 amu more than that of **2**, suggesting the presence of a monosaccharide moiety. A significant fragment using the high-resolution electron impact mode (HR-EIMS) appeared at 327.0868 corresponding to C_18_H_15_O_6_ (M^+^-glucose). A positive Molisch’s test reflected the glycosidic nature [26]. Absorption bands at 3451, and 1645 cm^−1^ were observed in the IR spectrum assigned to OH and C=O, respectively.

The detailed NMR spectral analyses for **3** were also very similar to that of **2,** particularly the aglycone that showed slight chemical shift differences due to the glycosylation site of the aglycone (i.e., left-hand side of the molecule). The main difference in the ^1^H NMR (Table 1) data between the two compounds was the downfield shift of H-5 and H-7 from δ_H_ 6.20 and δ_H_ 6.19 (**2**) to δ_H_ 6.51, representing two protons (**3**), respectively, accompanied by the downfield shift of H-4 from δ_H_ 2.87 in **2** to δ_H_ 2.92 in **3**. As expected, C-5 and C-7 in **3** were also downfield shifted by + 0.4 and + 1.4 ppm compared to those in the aglycone (**2**). Furthermore, the monosaccharide part was identified as glucose by the appearance of a doublet signal H-1″ at δ_H_ 4.99 with a large J value (7.2 Hz), indicating its β-configuration. The remaining proton and carbon signals of glucose were assigned carefully by ^13^C, DEPT-^13^C, and 2D NMR, and were in good agreement with those reported for glucose [27]

The above data confirmed that **3** is the glycoside of **2**. The glycosylation site was confirmed as C-6 by *J*_2&3_ bond correlations observed in the HMBC experiment from H-1″ to C-6; and H-5, H-7 to C-6. Other significant HMBC cross-peaks were observed from H-7 to C-1; H-3 to C-1 and C-4a; H-4 to C-3 and C8a; H-1′ to C-3′ and C-7′; H-7′ to C-3′, C-2′ and C-6′; and H-5′ to C-3′ and C-6′ (Figure 2). ECD spectral data of compound **3** were similar to those reported for the known dihydroisocoumarin derivative, feralolide (**2**) previously isolated from *A. ferox* [20]. The final structure of **3** was determined as 3,4 dihydro-6-glucopyranozyl-8-hydroxy-3-(2′-acetyl-3′,5′-dihydroxyphenyl)methyl-1H-[2]benzopyran-1-one, isolated for the first time from nature and given the name vacillanoside. Notably, a similar isocoumarin glycoside (feralolide 3′-*O*-glycosyl) from *A. hildebrandtii* [27] and *A. arborescens* [21] was previously reported, but with a different glycosylation position.

Compounds (**10**) and (**11**) were obtained as red amorphous powders. HR-ESI-MS of both compounds gave [M−H]^−^ at *m*/*z* 579.1509 (calcd 579.1508 for C_30_H_27_O_12_^−^) with 17 degrees of unsaturation. Thirty carbon resonances were clear in the ^13^C-NMR and were sorted in a DEPT experiment into two oxymethylenes (*δ*_C_ 64.5, C-11 and *δ*_C_ 64.4, C-6′), 16 methines, and 12 quaternary carbons. The HR-EI-MS for both compounds showed strong peaks at 256.0735 with a relative abundance of 100% calculated for C_15_H_12_O_4_ and assigned to the aloe-emodin anthrone aglycone [19]. UV and IR spectral data of both compounds showed close similarity; each showed a λ_max_ in the UV spectrum at 209, 244, 300, and 330 nm, and the IR spectrum showed bands at 3400, 1606, 1511, 1240, 1640, and 1720 cm^−1^, indicating the presence of hydroxyl groups, aromatic rings, and chelated carbonyl and conjugated ester carbonyl groups. Both compounds gave a positive Molisch’s test, reflecting their glycosidic nature [26].

The detailed NMR spectral analyses of **10** and **11** showed a resemblance to those of compounds **8** and **9**, identified as microdontin A and B and originally isolated from *A. microdonta* [23]. The similarities of **10** and **11** in their NMR spectra can be summarized as follows: Both were *C*-glycosides of an aloe-emodin-9-anthrone derivative with a glucose unit esterified with caffeic acid at C-6′. This structure was confirmed by ^1^H-NMR signals for aloe-emodin-9-anthrone observed as a pair of meta-coupled aromatic protons resonating at *δ*_H_ 6.80 (brs, H-2) and 6.97 (brs, H-4) in (**10**) and 6.77 ppm (d, *J* = 2.2 Hz, H-2) and 6.97 (d, *J* = 2.2 Hz, H-4) in (**11**). Moreover, a monosubstituted ring A in both **10** and **11** was verified by three ABC coupled aromatic protons assigned to H-5, H-6, and H-7 that appeared at *δ*_H_ 6.90 (d, *J* = 8.0 Hz), *δ*_H_ 7.39 (t, *J* = 8.0 Hz), and *δ*_H_ 6.69 (d, *J* = 8.0 Hz) in **10**, and at *δ*_H_ 7.02 (d, *J* = 7.9 Hz), *δ*_H_ 7.42 (br t, *J* = 7.9 Hz), and *δ*_H_ 6.81 (d, *J* = 7.9 Hz) in **11**. The remaining protons for the aloe-emodin-9-anthrone moiety resonated at *δ*_H_ 4.46 (br s, H-10) in **10** and 4.54 ppm (br s) in **11**, and the hydroxymethyl group at position 3 resonated at *δ*_H_ 4.62 (br s) and *δ*_H_ 4.60 (br s) and was interpreted for H_2_-11 in **10** and **11**, respectively. The downfield carbons in the ^13^C-NMR at *δ*_C_ 195.3 ppm for **10** and 195.4 ppm for **11** were assigned to C-9.

The sugar moiety for both (**10**) and (**11**) proved to be *β*-D-glucopyranosyl connected to the aglycone via a C–C bond (*C*-glycoside), similar to microdontins A and B (**8** and **9**). This structure is indicated by the chemical shift of the anomeric proton of the sugar with *β*-configuration at *δ*_H_ 3.30 (d, *J* = 9.5 Hz) in (**10**) and *δ*_H_ 3.29 (d, *J* = 9.4 Hz) in (**11**) correlated to C-1′ carbons at *δ*_C_ 85.8 and 85.9 ppm, respectively, in the HSQC (Heteronuclear Single-Quantum Correlation) experiment. The remaining sugar signals (H2′–H6′) were in complete agreement with the signals reported for microdontins A and B [23]. The glycosidation site in **10** and **11** was confirmed at C-10 by significant cross-peaks in the HMBC experiment from H-1′ to C-4a and C-5a, and from H-10 to C-4, C-4a, C-5, C-5a and C-1′. Additionally, the methylene protons at C-6′ were downfield shifted to *δ*_H_ 3.83 (dd, *J* = 11.7, 6.9 Hz, H-6′a) and *δ*_H_ 4.23 (br d, *J* = 11.7 Hz, H-6′b) compared to aloins A (**6**) and B (**7**), indicating esterification at C-6′. The downfield shift of C-6′ from *δ*_C_ 63.1 ppm in aloin B to *δ*_C_ 64.4 and 64.5 ppm in **10** and **11**, respectively, (around 1.5 ppm) further support C-6′ acylation. This result also agrees with HMBC correlations from H-6′ to C-9″ and from the trans-olefinic protons to C-1″, C-2″, and C-6″ (Figure 2).

A remarkable difference was observed in the aromatic region between microdontin A and B and **10** and **11**. The phenolic acid in microdontins A and B was identified as *p*-hydroxycinnamic acid, but in **10** and **11** was identified as a caffeic acid moiety. This finding was confirmed by three coupled aromatic protons at *δ*_H_ 6.83 (d, *J* = 8.2 Hz), *δ*_H_ 6.97 (br d, *J* = 8.2 Hz), and *δ*_H_ 7.09 (br s), forming an ABX system and assigned for H-5″, H-6″, and H-2″. The system was accompanied with a trans-olefinic system H-7” and H-8” [*δ*_H_ 7.33 (d, *J* = 15.9 Hz), and *δ*_H_ 6.06 (d, *J* = 15.9 Hz)] in **10** and [*δ*_H_ 7.35 (d, *J* = 15.9 Hz), and *δ*_H_ 6.08 (d, *J* = 15.9 Hz)] in **11**. These data matched data reported for caffeic acid [28]. The signal in ^13^C-NMR at *δ*_C_ 168.9 was assigned to C-9″ in the two compounds. HMBC correlations established further evidence for C-6′ where cross-peaks from H-6′ to C-9″ and from the trans-olefinic protons to C-1″, C-2″, and C-6″ were observed (Figure 2).

Overall, the above data prove that compounds **10** and **11** are diasteroisomers and derivatives of microdontins A and B (**8** and **9**). Isomer A or B was established in a NOESY experiment. Clear cross-peaks were observed from H-10 to H-4, H-5, and H-1′ in **11**, but not in **10**, confirming the *α* form in the latter. Furthermore, the α orientation of the glucose moiety at C-10 in **11** was confirmed by comparing its ECD spectrum with the related compounds, aloin B, microdontin B, and 10-hydroxy aloin B [29], proving that **11** is the *β* isomer and **10** the *α* isomer.

Trivial names, vacillantin A and B, were given to compounds **10** and **11**. Notably, for all of the isolated compounds with α orientation at H-10 (**6**, **8**, and **10**), the chemical shift of H-4 was more downfield than H-5, while the opposite occurred in the *β* equivalents (**7**, **9**, and **11**) Table 2. Moreover, during RP C_18_ HPLC separation using MeOH/H_2_O, the polarity of the *α* isomer was less than the polarity of the *β* form.

### 2.2. Antioxidant Activities Results

Antioxidant activity was evaluated by five different spectrophotometric methods, namely DPPH (2,2-diphenyl-1-picrylhydrazyl), ABTS (2,20-azinobis-(3-ethylbenzothiazoline-6-sulfonic acid), FRAP (Ferric reducing antioxidant power), superoxide, and nitric oxide radical-scavenging assays (Appendix A). All of the extracts displayed dose-dependent reducing activity. Mean percent scavenging ± SD, was measured in DPPH, ABTS, and FRAP assays at four concentrations (10, 20, 50, and 100 µg/mL) (Table 3), while for the superoxide and nitric oxide scavenging assays, it was calculated at 20, 40, 60, 80, and 100 µg/mL. The IC_50_ ± SD was determined for each assay.

The dichloromethane fraction demonstrated the highest radical scavenging activities at all concentrations (10, 20, 50, 100 µg/mL) (Table 3), with 81.90 ± 1.43% and 83.49 ± 3.09% scavenging at a concentration of 100 µg/mL, and IC_50_ of 22.14 ± 2.25 and 13.51 ± 2.33 in the DPPH and ABTS assays, respectively. These results are comparable to ascorbic acid (91.94 ± 0.92 and 90.51 ± 4.46% scavenging at 100 µg/mL concentration, and IC_50_ of 16.7 ± 1.96 and 10.56 ± 1.74 in the DPPH and ABTS assays, respectively).

The dichloromethane fraction also showed medium ferric reduction capability verified by a higher intensity of Perl’s Prussian blue color measured at 700 nm (1.44 ± 0.07% inhibition and IC_50_ 30.64 ± 2.46). In comparison, ascorbic acid showed 1.84 ± 0.11 ferric reduction capability, at the same concentration (100 mg/mL) and IC_50_ of 16.3 ± 1.82. The *n*-butanol extract exhibited 0.86 ± 0.08% activity (IC_50_ 30.64 ± 2.46), while the MeOH and the EtOAc extracts showed even weaker ferric reduction activities (IC_50_ of 94.83 ± 5.44 and 118.3 ± 5.8, respectively).

All tested samples produced concentration-dependent antioxidant effects in the superoxide scavenging assay (Table 4). Both CH_2_Cl_2_ and *n*-BuOH fractions displayed moderate free radical-scavenging, 87.90 ± 1.77% and 85.86 ± 2.26% (IC_50_ 32.92 ± 2.99 and 46.86 ± 5.1, respectively), compared to ascorbic acid (90.92 ± 1.70% and IC_50_ 7.09 ± 3.09).

Similarly, the CH_2_Cl_2_ fraction produced the highest inhibition activity among the tested samples in the nitric oxide scavenging assay (Table 5) with 80.03 ± 3.43% inhibition at the concentration of 100 µg/mL (IC_50_ 37.23 ± 3.72) compared to ascorbic acid (89.28 ± 2.02% inhibition, IC_50_ 22.37 ± 3.82). Conversely, MeOH and EtOAc fractions showed moderate to weak inhibitory actions in all assays.

The higher activity of both CH_2_Cl_2_ and *n*-BuOH is mainly due to the presence of a higher content of polyphenolic compounds including anthraquinones, flavonides, tannins, etc. in these extracts [30]. Antioxidant activity of leaf gel from *A. ferox*, as estimated using the FRAP assay, was attributed to its polyphenolic and alkaloid contents [30]. Furthermore, extracts from several *Aloe* species showed potent antioxidant potential in various in vitro assays including *A. barbadensis* [31], *A. arborescens* [32], *A. ferox* [33], *A. greatheadii* var. *davyana* [34], *A. harlana* [35], *A. saponaria* [36], *A. marlothii*, and *A. melanacantha* [37]. The activity was attributed to several phytochemical constituents such as loesin, aloeresin A, and aloesone [38].

Aloe-emodin, one of the main anthraquinone compounds isolated from *Aloe* spp., displayed strong antioxidant activity [38], revealed by its powerful reducing properties and the ability to inhibit the oxidation of linolenic acid [39]. In contrast, aloin, found in most *Aloe* plants, exhibited very similar properties, inhibiting lipid peroxidation in the cerebral cortex by inactivation of Fe(II)-dependent ascorbate [40].

In addition, the anthrone C-glycoside microdentin A/B isolated from the leaf latex of *A. schelpei* displayed stronger antioxidant activity compared to aloinoside A/B and aloin A/B using vitamin C as a standard [41].

## 3. Materials and Methods

### 3.1. Apparatus and Chemicals

Silica gel (Merck 60 A, 230–400 mesh ASTM, Darmstadt, Germany) was used for column chromatography. Normal and reversed phase silica gel (Merck, Darmstadt, Germany) were used for thin-layer chromatography (TLC). Anthraquinones were detected using a 254/366 nm UV lamp, followed by exposure to concentrated ammonia vapors or by spraying with 10% alcoholic KOH or NaOH. Additionally, compounds were visualized spraying with 15% H_2_SO_4_/ethanol, followed by heating.

HPLC analysis was performed on a Prominence Shimadzu LC Solution (Shimadzu Corporation, Kyoto, Japan) system with an InertSustain^®^ C_18_ analytical column (250 × 10 mm i.d.; 5 μm particle size) and a GL Sciences C_18_ preparative column (250 × 20 mm i.d.; 5 μm particle size) protected by a Waters Novapack RP C_18_ column guard. A binary LC-10AD pump, inline degasser, auto-sampler, and HP-1040A photodiode array detector coupled to an HP-85 personal computer were used for the analysis. UV–Vis spectra were recorded in the 200–700 nm range.

NMR spectroscopy was performed using deuterated solvents in an UltraShield Plus 500 (Bruker, Billerica, MA, USA) spectrometer operating at 500 MHz for ^1^H and 125 MHz for ^13^C. Some measurements used a Bruker AV-700 MHz NMR spectrometer (Bruker, Billerica, MA, USA) operating at 700 MHz for ^1^H and 175 MHz for ^13^C at the College of Pharmacy, King Saud University. Chemical shift values are reported in δ (ppm) relative to an internal standard (TMS) or residual solvent peak, and coupling constants (*J*) are reported in Hertz (Hz). The standard Bruker pulse program was used for the two-dimensional NMR analyses (COSY, HSQC, HMBC, and NOESY). HRMS was conducted by direct injection using a Thermo Scientific UPLC RS Ultimate 3000 Q Exactive Hybrid Quadrupole-Orbitrap Mass Spectrometer (company, city, country) (Mundelein, Illinois 60060 USA) combined with high-performance quadrupole precursor selection with high resolution, accurate-mass (HR/AM) Orbitrap™ detection. Direct infusion of isocratic elution was done using CH_3_CN/MeOH (7:3) as a solvent system with 0.1% formic acid. The experiment time was run for 1 min using nitrogen as the supplementary gas. A scan range from 160–1500 *m/z* was used. Detection was performed in both positive and negative modes, separately. The instruments were located at Prince Sattam Bin Abdulaziz University, College of Pharmacy. In addition, accurate mass determination was also achieved with a JEOL JMS-700 High-Resolution Mass Spectrophotometer (JEOL USA Inc., Peabody, MA, USA). The electron impact mode with an ionization energy of 70 eV was adopted. A direct probe was used with the following temperature ramp settings: Initial temperature of 50 °C; increasing by 32 °C/min, reaching a final temperature of 350 °C; resolution was adjusted to 10 k. IR spectrum was acquired using a JASCO 320-A spectrometer (JASCO International Co., Ltd., Easton, MD, USA). ECD analysis was performed on a J-815 CD spectrometer (JASCO INTERNATIONAL CO., LTD., Easton, MD, USA). A BioTek PowerWave 200 Microplate Spectrophotometer (BioTek, Winooski, VT, USA) and a PerkinElmer EnVision Multilabel Microplate Reader (PerkinElmer EnVision, Waltham, MA, USA) were used to monitor the antioxidant capacity of the tested samples. Reagents, chemicals, and solvents were analytical grade, purchased from Sigma-Aldrich (St. Louis, MO, USA), Loba Chemie Pvt. Ltd. (Mumbai, India), and SD Fine Chem. Ltd. (Mumbai, India).

### 3.2. Plant Material

The leaves of *A. vacillans* Forssk were collected in February 2018 in Mahayil Asir, in the southwestern region of Saudi Arabia (latitude: 18°13′0.4692″ N and longitude: 42°30′13.5540″ E). The specimens were kindly identified by Dr Raja Krishnan, Botany and Microbiology Department at the College of Science, King Saud University, Riyadh, Saudi Arabia. A voucher specimen (#11965) was submitted to the herbarium of the College of Science, King Saud University.

### 3.3. Extraction and Isolation

The succulent leaves of *A. vacillans* (4 kg) were chopped into small pieces and extracted in 70% hot methanol several times until exhaustion. The pooled extracts were then concentrated in a rotary evaporator to obtain a dark semi-solid residue (180 g). Total methanolic extract (MeOH) was dispersed in 300 mL of distilled water and successively partitioned with dichloromethane (CH_2_Cl_2_), ethyl acetate (EtOAc), and *n*-butanol (*n*-BuOH). Organic fractions were filtered over anhydrous sodium sulfate and evaporated to dryness to yield fractions A (DCM, 4.0 g), B (EtOAc, 8.5 g), C (*n*-BuOH, 70 g), and D (aqueous fraction, 90 g). The fractions were monitored on normal and RP C_18_ TLC using different solvent systems: *n*-hexane/EtOAc, CH_2_Cl_2_/MeOH, and CH_2_Cl_2_/MeOH/H_2_O at different ratios. The CH_2_Cl_2_ and EtOAc fractions were the richest in secondary metabolites including anthraquinones, triterpenes, and sterols; based on these results, these extracts were chosen for further chromatographic investigation.

Part of the CH_2_Cl_2_ fraction (3.5 g) was chromatographed on a silica gel column (100 × 4 cm, 358 g silica) using the *n*-hexane/EtOAc solvent system, followed by EtOAc/MeOH in gradient elution mode, yielding 251 fractions. Similar fractions, monitored on Kiesel gel 60 F_254_ TLC, were combined to give six main fractions (A–F). Direct crystallization of fraction C, eluted by 40% EtOAc/*n*-hexane, afforded compound **1**, aloe-emodin, while compound **2**, dihydroisocoumarin, was recovered from fraction E after sub-column treatment, on a silica gel column, using a CH_2_Cl_2_/MeOH solvent system in gradient elution mode.

The EtOAc extract (8 g) was applied on top of a silica gel column (100 × 4 cm, 500 g silica) and eluted with a CH_2_Cl_2_/MeOH mixture of increasing polarity. Eighty sub-fractions were collected and monitored on F_254_ TLC using *n*-hexane/EtOAc, and CHCl_3_/MeOH at different concentrations as well as on RP C_18_ TLC using MeOH/H_2_O at different ratios. Based on the results obtained from TLC, similar fractions were combined to yield seven main fractions (I–VII). Two fractions were chosen for further purification by reversed-phase C_18_-HPLC in a gradient elution mode starting with MeOH-H_2_O (60:40). Fraction II (130 mg) was eluted over 70 min (flow rate, 2 mL/min) to afford pure compounds **3** (1 mg, R_t_ = 17.88 min, 69.4% MeOH/H_2_O), **4** (15 mg, R_t_ = 21.25 min, 71.9% MeOH/H_2_O), **5** (12 mg, R_t_ = 23.12 min, 73.1% MeOH/H_2_O), **6** (12 mg, R_t_ = 30.97 min, 78.9% MeOH/H_2_O), **7** (9 mg, R_t_ = 32.42 min, 79.9% MeOH/H_2_O), **8** (7 mg, R_t_ = 38.95 min, 84.7% MeOH/H_2_O), and **9** (13 mg, R_t_ = 39.82 min, 85.3% MeOH/H_2_O). Similarly, fraction III (300 mg) was purified over 70 min to provide **10** (6 mg, R_t_ = 40.1 min, 85.5% MeOH/H_2_O), **11** (8 mg, R_t_ = 40.95 min, 86.1% MeOH/H_2_O), and **12** (3 mg, R_t_ = 47.86 min, 91.2% MeOH/H_2_O) (Scheme 1).

### 3.4. Antioxidant Activity

#### 3.4.1. DPPH Radical Scavenging Assay

The antioxidant activity of the extracts and fractions was determined using DPPH (2,2-diphenyl-1-picrylhydrazyl) based on the method described by [42]. Absorbance was determined after 30 min at 520 nm, and percentage inhibition was obtained with the following the equation:Scavenging (%) = A_0_ − A_t_/A_0_ × 100(1)
where A_t_ is the absorbance of the extract and A_0_ is the absorbance of the control.

#### 3.4.2. ABTS Radical Cation Scavenging Assay

The assay was performed following the procedure described by [43]. The ability of samples to reduce the ABTS free radical (2,20-azinobis-(3-ethylbenzothiazoline-6-sulfonic acid) was also estimated using the above formula.

#### 3.4.3. Reducing Power Assay

The reducing power of the extracts was determined using the method adapted by [44]. The antioxidant method (i.e., FRAP) is based on the capability of a test sample to reduce ferric ions (Fe^3+^) to ferrous ions (Fe^2+^) by electron donation.

#### 3.4.4. Superoxide Radical Anion Scavenging Assay

Superoxide anion radical scavenging activity was assessed as previously described [45] with slight modification. Superoxide radicals were created by oxidation of NADH in a PMS-NADH system, and antioxidant activity was measured by the extent to which the extract and fractions of *A. vacillans* reduced nitro blue tetrazolium (NBT). The percentage of superoxide radical scavenging was also calculated using the above formula.

#### 3.4.5. Nitric Oxide Radical Scavenging Assay

The assay was performed as previously described [46]. The free radical scavenging activity of the extract and fractions was determined by evaluating the % inhibition of the nitrite ions generated from the interaction of nitric oxide with oxygen using the same equation above-mentioned.

#### 3.4.6. Spectral Data of the New Compounds

##### Vacillanoside (**3**)

White amorphous powder (1 mg); [α]^23^_D_ − 56.2° (*c* 0.1, MeOH; UV λ_max_ MeOH (log ε) nm: 211 (4.46), 274 (4.22), 306 (3.99); IR (KBr) *v*_max_ 3451, 1645, 1631, 1608, 1595, 1054, 1032, and 1016 cm^−1^; ^1^H and ^13^C NMR (500, 125 MHz, in CD_3_OD) HR-ESI-MS: *m*/*z* 505.1353 [M−H]^+^ (calcd 505.1346 for C_24_H_25_O_12_), *m*/*z* 507.1500 [M+H]^+^ (calcd 507.1503 for C_24_H_26_O_12_+H), *m*/*z* 529.1320 [M+Na]^+^ (calcd 529.1322 for C_24_H_26_O_12_Na), *m*/*z* 545.1069 [M+K]^+^ (calcd 545.1061 for C_24_H_26_O_12_K).

##### Vacillantin A (**10**)

Red amorphous powder (6 mg); [α]^23^_D_ + 16.8° (*c* 0.05, MeOH); UV λ_max_ MeOH (log ε) nm: 209 (4.63), 244 (4.42), 300 (3.92), 330 (3.61); IR (KBr) *v*_max_ 3400, 1720, 1640, 1606, 1511, 1240 cm^−1^; ^1^H and ^13^C NMR (see Table 2); HR-ESI-MS: *m*/*z* 579.1509 [M−H]^+^ (calcd 579.1503 for C_30_H_27_O_12_).

##### Vacillantin B (**11**)

Red amorphous powder (8 mg); [α]^23^_D_ − 4.7° (*c* 0.05, MeOH); UV λ_max_ MeOH (log ε) nm: 209 (4.63), 244 (4.42), 300 (3.92), 330 (3.61); IR (KBr) *v*_max_ 3400, 1720, 1640, 1606, 1511, 1240 cm^−1^; ^1^H and ^13^C NMR (see Table 2); HR-ESI-MS: *m*/*z* 579.1509 [M−H]^+^ (calcd 579.1503 for C_30_H_27_O_12_), *m*/*z* 581.1651 [M+H]^+^ (calcd 581.1659 for C_30_H_28_O_12_ + H), *m*/*z* 603.1469 [M+Na]^+^ (calcd 603.1478 for C_30_H_28_O_12_ + Na).

### 3.5. Statistical Analysis

Analysis of variance (ANOVA) was used to evaluate significance differences, followed by the Student’s *t*-test. Data were expressed as mean ± SD, and the difference was considered significant at *p* < 0.05 compared to the control. All statistical calculations used OriginLab software (version 8, Northampton, Massachusetts, USA) and Microsoft Excel.

## 4. Conclusions

In summary, a new dihydroisocoumarin derivative, vacillanoside (**3**), 3,4 dihydro-6 glucopyranozyl-8-hydroxy-3-(2′-acetyl-3′,5′-dihydroxyphenyl)-methyl-1H-[2]benzopyran-1-one (**6**), and two new anthraquinone derivatives, vacillantins A and B (**10** and **11**) were isolated from the leaves of *A. vacillans* (Asphodelaceae) together with nine known compounds (**1**, **2**, **4****–****9**, and **12**). The structures of these compounds were elucidated through extensive spectroscopic analyses. The total alcohol extract and different fractions were tested for their antioxidant activities in five spectrophotometric assays. The dichloromethane fraction exhibited promising free radical scavenging activity in most of the assays. Our findings add new information to the literature on the structural diversity and pharmacological activities of *Aloe* species. Our results suggest *A. vacillans* as a potential source of secondary metabolites with pharmacological and industrial importance. Moreover, these results advocate further investigation of the remaining fractions with the aim of isolating bioactive compounds exhibiting interesting biological capacities.

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
