# Peer review of "Vacillantins A and B, New Anthrone C-glycosides, and a New Dihydroisocoumarin Glucoside from Aloe vacillans and Its Antioxidant Activities"

_plants, 2020, doi:10.3390/plants9121632_

Round 1

Reviewer 1 Report

It is interesting to determine the new anthron C-glycoside derivatives. Authors also measured the antioxidant avidities of the compounds using common methods. However, a few new derivatives of the compounds from the leaf extracts were found. The authors discussed nothing about the antioxidant activities.

Line 27 on page 1. Authors mentioned “support the potential use” but there is no clear explanation in the manuscript.

Line 216 on page 7. The results of antioxidant activities were using organic solvents. How did the authors conclude the results which apply to human beings?

Line 247 on page 9. There is no discussion why n-BuOH results are higher than others.                                                    

English need to be checked by a native scientific speaker.

Examples:

Line 21 on page 1. “Leaves were extracted” in the Abstract. This should be “The compounds were extracted or Leaves were used to extract the compounds”?

Line 36 on page 1. “several countries” in the Introduction is ambiguous expression.

And many others.

Author Response

Reviewer’s comments 

Dear Doctor

Thank you very much for your kind effort for reviewing our manuscript titled " Vacillantins A and B, new anthrone C-glycosides, and a new dihydroisocoumarin glucoside from Aloe vacillans and its antioxidant activities” we considered all Comments and send our reply to each point as shown below.

The authors discussed nothing about the antioxidant activities. 

Detailed discussion of the antioxidant activity results is mentioned in page 8, under section 2. (Results and discussion) and subsection 2.2 (Results of antioxidant activities). The paragraph was rephrased and additional information was added. Line 219-260 in yellow mark.  

Comment 1  

Line 27 on page 1. Authors mentioned “support the potential use” but there is no clear explanation in the manuscript. 

Response 

Since the results obtained showed promising radical scavenging activities, specially the dichloromethane extract which displayed the highest activities at all concentrations (Table 3) in DPPH, ABTS and nitric oxide assays compared to ascorbic acid. Detailed interpretation of the obtained results is found in p 8, line 225-260

Comment 2 

Line 216 on page 7. The results of antioxidant activities were using organic solvents. How did the authors conclude the results which apply to human beings? 

Response 

The plant originally was extracted in hydroalcohol, but in case of approving its use as a supplementary antioxidant medication, it can be extracted using the appropriate organic solvent. The solvent will be evaporated to produce the dried extract which then can be formulated in a suitable pharmaceutical dosage form. 

Comment 3 

Line 247 on page 9. There is no discussion why n-BuOH results are higher than others. 

Response 

Perhaps the significant antioxidant activity of n-butanol extract is mainly due to the presence of polyphenolic compounds such as flavonoids, tannins and anthraquinone glycosides which play an important and significant role in free radical scavenging activities. Plz, check page  8, line 245-246. Please kindly See the following references 

Loots, D.T.; van der Westhuizen, F.H.; Botes, L. Aloe ferox leaf gel phytochemical content, antioxidant capacity, and possible health benefits. J. Agric. Food Chem. 2007, 55, 6891–6896.

Pandey, K.B. and Rizvi, S.I. , Plant polyphenols as dietary antioxidants in human health and disease. Oxid Med Cell Longev. 2009 Nov-Dec; 2(5): 270–278 

Comment 4 

English need to be checked by a native scientific speaker. 

 Response 

English and grammar editing was conducted. a Certificate issued by MDPI is attached below

Comment 5

Line 21 on page 1. “Leaves were extracted” in the Abstract. This should be “The compounds were extracted or Leaves were used to extract the compounds”?

Response

The sentence changed into The leaves were used to extract compounds with different solvents.

Comment 6

Line 36 on page 1. “several countries” in the Introduction is ambiguous expression.

Response

The sentence changed to Aloe plants are used as traditional medicines and dietary supplements in several countries including Egypt, China and India 

Reviewer 2 Report

I think that this work is very interesting but there are some points to clarify: 

line 46 format the text

line 231 like results ...? it is not clear this sentence

results: in my opinion you have to add IC50 values of different scavenging activities and compare this value to the IC50 value of the standard used in each test..

how many concentration of ascorbic acid do you test? i think you have to add calibration curve of your standard to validate youe method..

why not do you test also single identified compound? i think coud be interesting understand if there are a sinergy between different compounds or only one of them affect activity of the extract.. 

materials and methods section: could you describe briefly how you have performed different antioxidant assays?

Author Response

Reviewer 2 

Dear doctor

Thank you very much for your kind effort for reviewing our manuscript titled " Vacillantins A and B, new anthrone C-glycosides, and a new dihydroisocoumarin glucoside from Aloe vacillans and its antioxidant activities” we considered all Comments and send our reply to each point as shown below.

Comment 1 

line 46 format the text 

Response 

Corrected 

Comment 2 

line 231 like results ...? it is not clear this sentence 

Response 

The sentence was changed to the following: 

All tested samples produced concentration-dependent antioxidant effects in the superoxide scavenging assays (Table 4) at concentration of 100 µg/mL. Both CH2Cl2 and n-BuOH fractions displayed strong free radical-scavenging, 87.90 ± 1.77 % and 85.86 ± 2.26 % (IC50 32.92 ± 2.99 and 46.86 ± 5.1, respectively), compared to ascorbic acid (90.92 ± 1.70 % and IC50 7.09 ± 3.09).  

Comment 3 

Results: in my opinion you have to add IC50 values of different scavenging activities and compare this value to the IC50 value of the standard used in each test. 

Response 

IC50 values were added in Tables 3, 4 and 5 and compared to the IC50 of the standard used (ascorbic acid). 

Comment 4 

How many concentration of ascorbic acid do you test? i think you have to add calibration curve of your standard to validate your method. 

Response 

Five concentrations.

Calibration curves of samples and standard for the five used methods were added to the supplementary data.

Comment 5 

why not do you test also single identified compound? i think coud be interesting understand if there are a sinergy between different compounds or only one of them affect activity of the extract.  

Response 

A current study is undergoing to evaluate the antioxidants activities for the isolated compounds as well as standardization of the titled plant by different chromatographic techniques. 

Comment 6 

materials and methods section: could you describe briefly how you have performed different antioxidant assays? 

 Response 

To ovoid the repetition of information from other references which might increase the percentage of similarity (plagiarism) a detailed description of the procedure for each method used is added as supplementary data.

Round 2

Reviewer 1 Report

I think that this is acceptable.

Reviewer 2 Report

Thank you for modifications. Now paper is available for pubblication in my opinion.